# Combining Ground Penetrating Radar and Frequency Domain Electromagnetic Surveys to Characterize the Structure of the Calderone Glacieret (Gran Sasso d'Italia, Italy)

**Mirko Pavoni** [1,*] **, Jacopo Boaga** [1] **, Alberto Carrera** [2] **, Stefano Urbini** [3] **, Fabrizio de Blasi** [4,5] **and Jacopo Gabrieli** [4,5]

1    Department of Geosciences, University of Padova, 35122 Padua, Italy; jacopo.boaga@unipd.it
2    Department of Agronomy, Food, Natural Resources, Animals and Environment, University of Padova, 35122 Legnaro, Italy; alberto.carrera@phd.unipd.it
3    Istituto Nazionale di Geofisica e Vulcanologia, Dipartimento Ambiente, 35137 Rome, Italy; stefano.urbini@ingv.it
4    Institute of Polar Sciences, National Research Council (CNR-ISP), 30100 Venice, Italy; fabrizio.deblasi@unive.it (F.d.B.); gabrieli@unive.it (J.G.)
5    Department of Environmental Sciences, Informatics and Statistics, Ca' Foscari University of Venice, 30123 Venice, Italy
*    Correspondence: mirko.pavoni@phd.unipd.it

**Abstract:** Ice is a rich reservoir of past climate information, and the well-documented increasing rate of glacier retreat represents a great loss for paleoclimate studies. In this framework, the Ice Memory project aims to extract and analyze ice cores from glacier regions worldwide and store them in Antarctica as a heritage record for future generations of scientists. Ice coring projects usually require a focused geophysical investigation, often based on Ground Penetrating Radar (GPR) prospecting to assess the most suitable drilling positions. As a novel approach in the Calderone Glacieret, we integrated the GPR method with Frequency Domain Electromagnetic (FDEM) surveys, a technique not commonly applied in the glacial environment. We used a separated-coils FDEM instrument to characterize the glacieret structure. The acquired FDEM datasets were inverted and compared to the GPR data and borehole information. The results demonstrated the capability of the FDEM technique to define the structure of the glacieret correctly; therefore, the potential to be applied in frozen subsoil environments. This opens new perspectives for the use of the FDEM technique to characterize periglacial environments, such as rock glaciers, where the coarse-blocky surface hinders data acquisition and enhances the problem of signal scattering.

**Keywords:** FDEM; EMI; GPR; Calderone Glacieret; cryosphere; environmental geophysics

## 1. Introduction

The Calderone Glacieret is one of the southernmost ice bodies in Europe and the only one left in the Apennine mountains [1]. Unlike glaciers, which move downward under the buoyancy of their own weight, a glacieret is a snow and ice structure with no recorded movement during the last twenty years. Nevertheless, like many alpine glaciers [2], the Calderone ice body has been in a retirement phase since the beginning of the 20th century [3]. This trend, connected to an increase in average annual air temperatures [4], has shown a clear acceleration since the 1960s [5–7].

Glaciers' and ice bodies' retirement are an important proxy of the climate change rate [8], but at the same time, it represents a serious loss of data for paleoclimatic studies. Geochemical analysis of ice samples extracted from glaciers and ice bodies allows the reconstruction of past climate and temperature trends. [9]. To save this important natural database, the international project 'Ice Memory' has been created. The focus of this project, recognized by UNESCO, is to collect and store ice samples from glaciers and subsoil ice

bodies that could disappear or dramatically retreat soon due to global warming. The extracted ice cores will be moved to Antarctica, where they will represent a precious paleo-climatic archive accessible to future generations of scientists. Since 2016, the international Ice Memory team has collected ice cores from glaciers worldwide. High-altitude glacier field campaigns were carried out in Europe, South America and Asia. In the Andes, Caucasus and Tibetan plateau, the ice cores were extracted respectively from Illimani, Elbrus and Belukha glaciers. In the Alps, the ice samples were collected on Col du Dome, Corbassiere and Gorner glaciers. Recently, the Italian Ice Memory team (composed of the Institute of Polar Science of the Italian National Council of Research ISP-CNR and the Ca' Foscari University of Venice) has planned to extract an ice core from the last remaining ice body in the Apennines: the Calderone Glacieret.

Choosing the position of the ice core drilling is the first challenge of each extraction campaign. For this reason, preliminary geophysical investigations were carried out to define the main subsurface morphologies, the ice body thickness and internal layering. The GPR method is historically and commonly used with success in glacier environment characterization [10–13]. Electric Resistivity Tomography (ERT) and seismic methods may be applied, but the Calderone Glacieret coring operation was scheduled to be at the end of April 2022, while the preliminary geophysical surveys were planned to be in the middle of March 2022. The presence of several meters of snow cover limited the use of ERT and active seismic methods. Therefore, we decided to combine the GPR technique with frequency domain electromagnetic prospecting (FDEM). This technique has a long history in hydrogeological, archeological and agricultural studies [14], but it has been rarely applied in glacial environments. In the Calderone site, GPR and FDEM data were acquired along two survey lines: one longitudinal and one orthogonal to the old ice body flow. Here we compare the results of the two techniques to test the potential of the FDEM method to characterize the subsurface containing an ice-rich layer. Since the required investigation depth was several tens of meters, we adopted a separated-coils FDEM probe (CMD-DUO, GF-Instruments). Due to the relatively low frequency of the transmitted signal and the wide separation of the coils, the device was able to reach the target depth.

Based on the glacieret models obtained from the results of GPR and FDEM measurements, the position for the core extraction was chosen, and the borehole was successfully realized on April 2022.

## 2. Site Description

The Calderone Glacieret is in the Abruzzo region (Central Italy—blue circle in Figure 1A), in the massif of the 'Gran Sasso d'Italia'. It is located at an altitude ranging between 2650 and 2850 m above sea level (a.s.l.), on the northern slope of the Corno Grande peak, the highest summit of the Apennine mountains (2912 m a.s.l.). From a geological point of view, the Corno Grande is composed entirely of a calcareous succession of the Triassic platform [15]. The Calderone Glacieret was able to survive below the limit of perennial snows thanks to the steep walls of a northeast-facing circus [5]. Furthermore, thanks to the northeastern exposition and the steep rock walls that intercept the winter precipitation coming from eastern Europe, a thick snow cover is ensured every winter [4]. During summer, the ice bodies are entirely covered by a layer of a few meters of calcareous debris which acts as a thermal insulator, protecting the underlying ice layers from solar radiation and preventing them from melting. Nevertheless, the Calderone Glacieret is probably in a transition phase to a periglacial form (e.g., rock glacier).

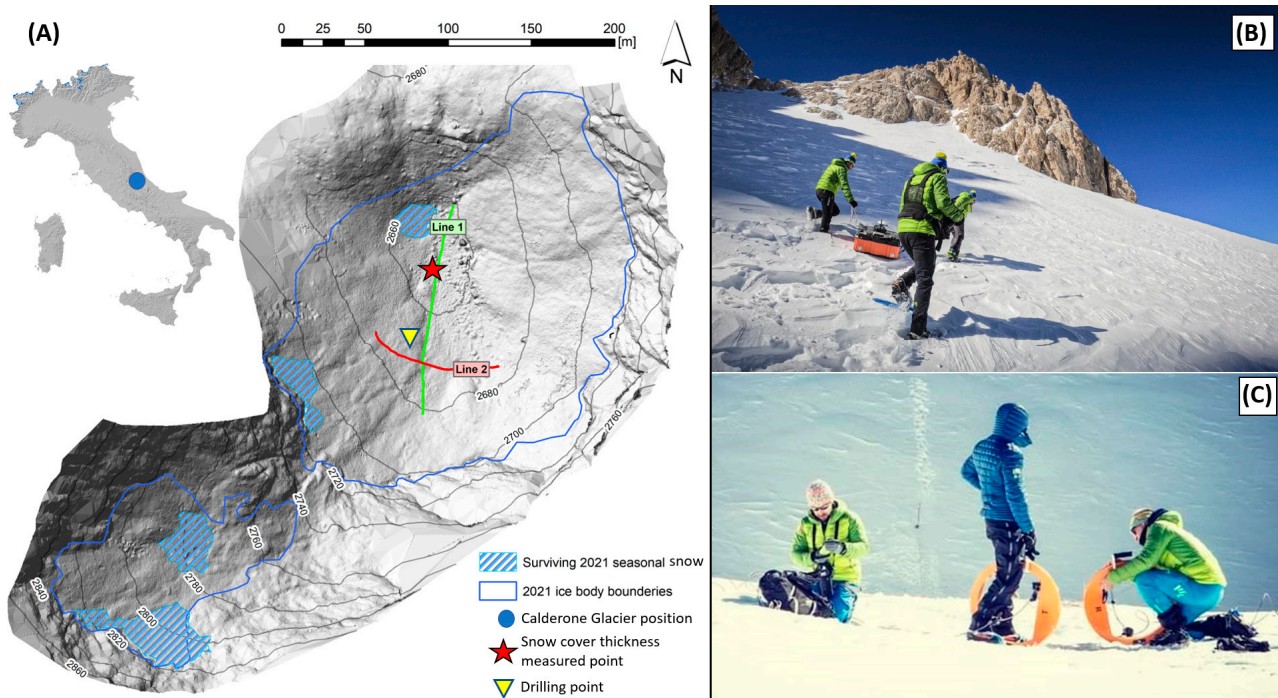

**Figure 1.** (**A**) Position of the Calderone Glacieret (blue circle) in Central Italy (EU-DEM v1.1—Copernicus Land Monitoring Service) and the location of the survey lines performed with (**B**) GPR and (**C**) FDEM methods. The hillshade raster from photogrammetric DTM, survey Line 1 (green line) in Figure 1A, is 135 m long and longitudinal to the development of Calderone Glacieret; Line 2 (red line) is 85 m long, and it is orthogonal to the development of Calderone Glacieret.

Radiometric dating techniques have been performed on the glacial deposits downstream and on the threshold of the Calderone circus, confirming that during the Holocene, the glacier had various phases of expansion and retreat [16]. According to these measurements, the last phase of expansion took place during the Little Ice Age, while the retreat phase has been well-documented since the early 1900s. Marinelli and Ricci [3] estimated that the Calderone ice body covered an area of 0.07 km$^2$ at the beginning of the 20th century. Tonini [5] defined its reduced area as 0.06 km$^2$ in the 1960s, and in 1990 the surface decreased by a further 20% [6]. The glacieret was almost entirely covered by a few meters of debris in the 1990s [7], and since the summer of 2000, was split into two different ice bodies (see Figure 1A). In 2015, GPR measurements estimated a maximum thickness of 26 m for the ice-rich layer in the northernmost ice body [8].

In March 2022, GPR (Figure 1B) and FDEM investigation lines (Figure 1C) were acquired to define the maximum thickness of the ice-rich layer. Two acquisition lines were measured with both geophysical techniques, Line 1 (green line in Figure 1A) and Line 2 (red line in Figure 1A). The first line is longitudinal to the development of the glacieret, practically on the same orientation as those performed by Pecci et al. [17] and Monaco and Scozzafava [18], while the second line has an orthogonal orientation.

## 3. Methods

### 3.1. Ground Penetrating Radar (GPR)

A glacial environment represents a very appropriate context for GPR applications since the dielectric properties of ice and snow lead to a low attenuation of the transmitted signal [10]. Pure ice has a relatively low dielectric constant which does not attenuate the high-frequency radar signal (in the order of MHz) transmitted by the probe. Furthermore, the thickness of the ice layer can be precisely estimated since the interface with the underly-

ing bedrock (which, on the contrary, has a relatively high dielectric constant) is highlighted by a clear reflection in the acquired radargram [19,20].

In the Calderone Glacieret survey, GPR measurements were collected on the surface of the snow cover using a monostatic digital antenna of 200 MHz (GSSI Sir4000 instrument—see Figure 1B). Table 1 shows the main acquisition parameters of the GPR survey. All the measurements were georeferenced with a Trimble R9s GNSS receiver in RTK configuration. Reflection arrival times were converted into depth using an averaged electromagnetic wave speed of 0.201 m/ns and 0.1682 m/ns for the snow cover and the ice layer, respectively. These values were calculated by taking an average of the hyperbola diffractions, where the medium separations emerged clearly. Data processing included commonly applied techniques such as vertical and horizontal bandpass filters, deconvolution, gain equalization and migration. These were performed using ReflexW software (Sandmeier geophysical research).

**Table 1.** GPR acquisition parameters used during the measurements performed on the Calderone Glacieret survey in March 2022.

| Investigation Range (ns) | Samples (Points) | Simple for Second | Dynamic (Bit) |
|---|---|---|---|
| 400 | 1024 | 40 | 32 |

### 3.2. Frequency Domain Electromagnetic (FDEM)

The FDEM method applies Maxwell's equations to estimate the electrical conductivity of the subsoil under investigation [21] without the need for a galvanic contact between the device and the ground surface. FDEM instruments have a transmitter coil (*Tx*) where an alternating current flow with a fixed frequency (*f*), which induces a primary magnetic field (*Hp*) with the same frequency as *f*. *Hp* propagates through the subsoil and induces secondary electrical currents. The latter, in turn, generates a secondary electromagnetic field (*Hs*) which is measured by the receiver coil (*Rx*). The ratio between *Hs/Hp* is a complex number. From the real part of the number (quadrature *Q*), the apparent electrical conductivity ($\sigma_a$) of the subsoil can be calculated, as shown in Equation (1):

$$\sigma_a = \frac{4}{\omega \mu_0 s} Q \tag{1}$$

where $\omega$ is the angular frequency ($\omega = 2\pi f$) of the transmitted signal, *s* is the separation distance of the two coils (*Tx* and *Rx*) and $\mu_0$ is the magnetic permeability of free space (considering that most of the subsoils are practically non-magnetic, McLachlan et al., 2021 [21]). This relationship is true only if the Low Induction Number (β) condition (LIN) is verified:

$$\beta = s\sqrt{\frac{2}{w\mu_0 \sigma}} \ll 1 \tag{2}$$

In a debris-covered glacieret environment, such as the Calderone site, the electrical conductivities are particularly low, and consequently, the LIN condition is always satisfied. The measured $\sigma_a$ is influenced by the relative contribution of the different layers that compose the subsurface, and the penetration depth of the measurements is linked to different factors: the separation *s* of the coils, their orientation (horizontal co-planar HCP or vertical co-planar VCP) and the transmitted frequency *f*. By using higher coil separations *s* or lower frequencies *f*, the measured apparent conductivity $\sigma_a$ will be more affected by the properties of the deeper layers. By considering fixed values of *s* and *f*, the HCP mode allows us to further increase the penetration depth of the survey with respect to the VCP mode (see Figure 2). In a debris-covered environment with very low electrical conductivities, the magnetic field decays rapidly and restricts the penetration depth [22]. This problem can be partially solved by using a lower frequency *f* and higher values of *s* [23]. Due to these limitations, in the Calderone Glacieret, we adopted a separated coils FDEM instrument,

the GF Instruments CMD-DUO (see Figures 1C and 2). The device has a relatively low transmitted frequency $f$ of 925 Hz and three large coil separations $s$ (10, 20 and 40 m). Moreover, both VCP and HCP modes can be implemented. This way, six $\sigma_a$ values can be obtained at each measured point (which is considered to be halfway between the two coils), defining an electrical conductivity profile from a few meters of depth to several tens of meters. Figure 2 shows the nominal depth range, suggested by the manufacturer (GF Instruments), which influences the measured apparent conductivities acquired with a CMD-DUO device.

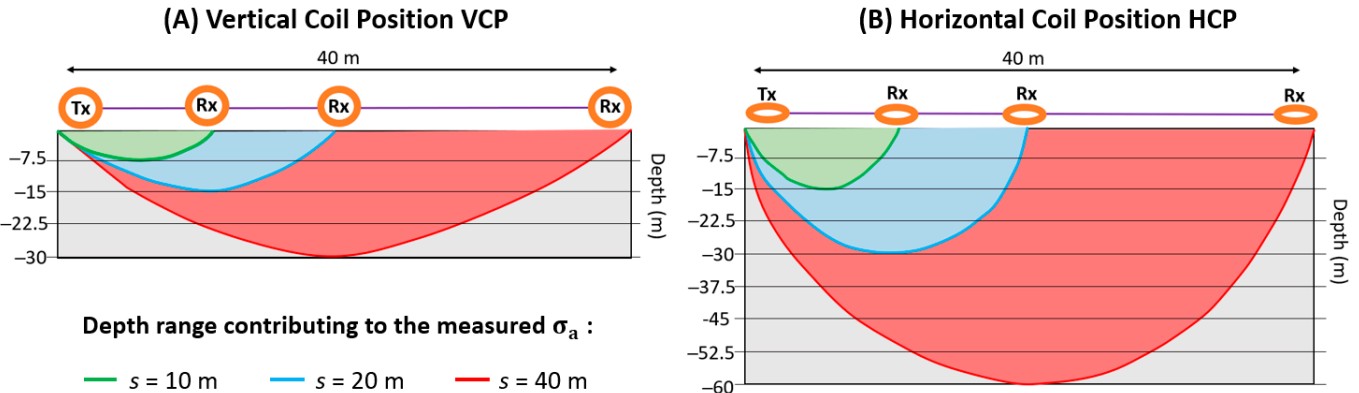

**Figure 2.** (**A**) Nominal depth range influencing the measured apparent conductivity $\sigma_a$ for the different CMD-DUO coil separation (s) considering the vertical coil orientation (VCP). (**B**) Depth range influencing the measured apparent conductivity $\sigma_a$ for the different CMD-DUO coil separation (s) considering the horizontal coil orientation (HCP).

The application of FDEM methods is limited by the instrumental resolution limit and the low electrical conductivity of the ice (in temperate glaciers ~$1 \times 10^{-3}$ mS/m [22]). In particular, GF Instruments FDEM devices (such as CMD-DUO) cannot estimate electrical conductivity variations below $1 \times 10^{-1}$ mS/m. Despite this, by considering the results in a relative way, it is possible to discriminate areas with lower conductivities (with possible ice fraction present in the subsurface) from areas with higher ones (without ice fraction). The technique has been recently used successfully to define the near subsurface structure of several alpine rock glaciers, e.g., [23,24].

*3.3. FDEM Forward and Inverse Modelling*

The forward and inverse FDEM modelling has been performed using the open-source Python-based software EMagPy [21]. To simulate the non-simplified response of the CMD-DUO survey, the Full Maxwell Solution has been used. The method considers the propagation of electromagnetic fields by conduction currents, valid only with frequencies $f < 10^5$ Hz (the CMD-DUO has a transmitted signal of 925 Hz). The forward modelling consists of the computation of the Hs/Hp ratio (see Equations (3) and (4)), once the characteristics of coil separation $s$, frequency $f$ of the transmitted signal, and thickness and electrical conductivities of a layered subsoil model are defined:

$$\left(\frac{H_S}{H_P}\right)_{VCP} = 1 - s^2 \int_0^\infty R_0 J_1(s\lambda)\lambda d\lambda \tag{3}$$

$$\left(\frac{H_S}{H_P}\right)_{HCP} = 1 - s^3 \int_0^\infty R_0 J_0(s\lambda)\lambda^2 d\lambda \tag{4}$$

where $J_0$ is a Bessel function of the zeroth order, $J_1$ is a Bessel function of the first order, and $R_0$ is the reflection factor, which is calculated using the thickness and electrical conductivities of the layers (for details, see [21]). Finally, Equation (1) allows us to find a synthetic dataset of $\sigma_a$ that would be measured by a FDEM device.

EMagPy was used to perform also the quasi-2D inversions of the acquired field datasets, generating inverted conductivity profiles at each measured point. The inverted profiles were then interpolated with the kriging method [25] to obtain a quasi-2D conductivity section (from now on, simply called "inverted conductivity sections" or "FDEM models"). Like all the geophysical methods, the inversion procedure is an iterative process aimed at minimizing the misalignment between the measured dataset of $\sigma_a$ and a synthetic dataset of $\sigma_a$ calculated using a forward model. Equation (5) shows the L2 norm objective function, which is minimized:

$$\frac{1}{N} \sum_{i=1}^{N} (d_i - F_i(m))^2 + \alpha \left( \frac{1}{M} \sum_{j}^{M-1} \left( \sigma_j - \sigma_{j+1} \right)^2 \right) \ \to \ min \tag{5}$$

In Equation (5), $N$ is the number of coil configurations (separations and orientations), $d$ contains the measured dataset of $\sigma_a$, $F(m)$ the apparent conductivities $\sigma_a$ calculated with the forward model, $M$ is the number of layers in the model, $\sigma_j$ is the conductivity of layer $j$, and $\alpha$ is the regularization parameter (defined with an L-Curve analysis) [26]. Among several techniques (see [21]), a straightforward solution to minimize Equation (5) is to use the Cumulative Sensitivity (CS) functions and the gradient-based optimization method of Gauss–Newton. For example, McNeil [27] proposed the CS functions, shown in Equations (6) and (7), to define the contribution of the subsurface layers to the measured apparent conductivities. The normalized sensitivities ($R$) for the two coil orientations are:

$$R_{VCP}(z) = \sqrt{(4z^2 + 1)} - 2z \tag{6}$$

$$R_{HCP}(z) = \frac{1}{\sqrt{(4z^2 + 1)}} \tag{7}$$

where $z$ is the depth normalized by the coil separation $s$. Finally, to evaluate the accuracy of the predicted conductivity models, EMagPy provides the Relative Root Mean Squared Error (RRMSE).

Data filtering was applied to facilitate the Calderone FDEM survey's inversion routine. In fact, as the datasets were acquired in challenging conditions, which involved walking with snowshoes on a steeply sloped snow cover of several meters (see Figure 1B,C), it was practically impossible to guarantee the perfect coils co-planar orientation and separation during the measurements. This likely contributed to the presence of anomalous measurements in the acquired datasets. For these reasons, preliminary data filtering was applied (e.g., Figure 3 presents the filtering of the Line 1 dataset collected with a coil separation of 40 m and HCP mode).

We first applied a detrend function (which removes both offsets and linear trends) to the raw datasets, and all the $\sigma_a$ values outside the confidence interval of Equation (8) have been deleted:

$$\mu - 2sd < \sigma_a < \mu + 2sd \tag{8}$$

where $\mu$ is the average $\sigma_a$ of the detrended dataset, and $sd$ is the standard deviation. The saved measurements have been then returned to their initial raw values and smoothed by interpolating with a 6th-grade polynomial function. Finally, to define the maximum depth of the models, sensitivity profiles of the measurements have been calculated. In the current work, the inverted FDEM models are limited to the depths where the normalized sensitivity of the measurements reaches approximately zero.

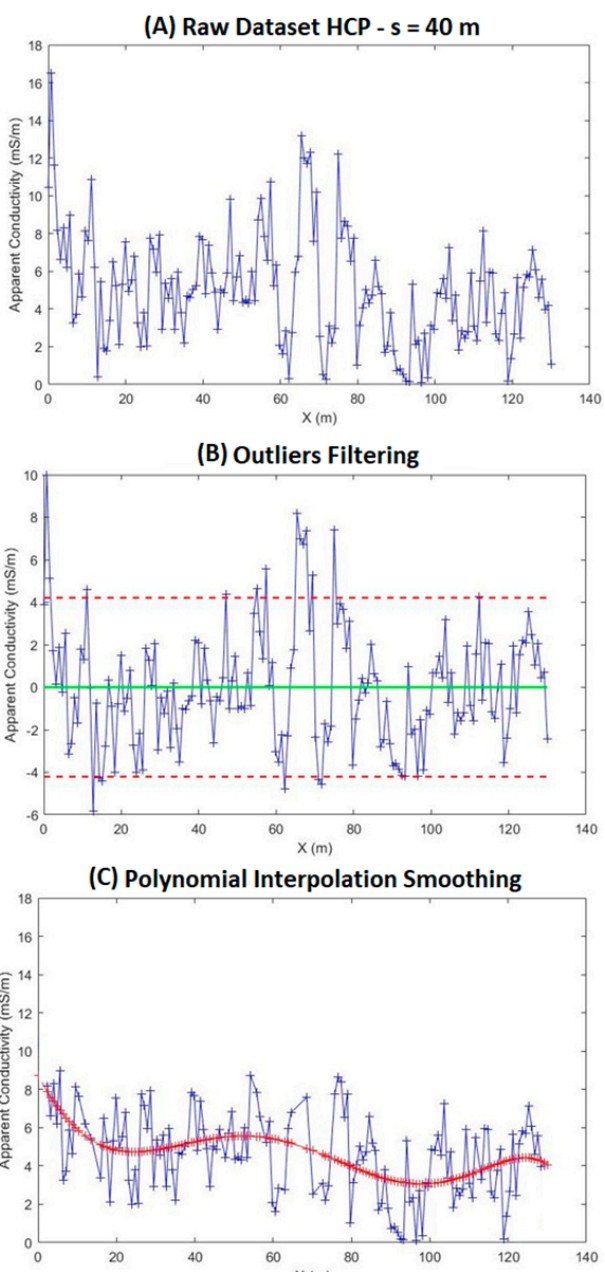

**Figure 3.** Example of the data filtering sequence applied to the raw measurements of Line 1. (**A**) Step 1: raw dataset acquired with coil separation of 40 m and horizontal coil orientation HCP. (**B**) Step 2: detrend function and filtering of the anomalous $\sigma_a$ values which are outside the confidence interval of $\mu - 2sd < \sigma_a < \mu + 2sd$ ($\mu$ is the average $\sigma_a$ of the detrended dataset, and $sd$ is the standard deviation). (**C**) Step 3: the saved measurements have been returned to their initial raw values and smoothed using a 6th-order polynomial function.

## 4. Results

*4.1. GPR Results*

In Figure 4, the post-processing results of the GPR measurements are presented. In both profiles, the snow layer is characterized by low attenuation of the transmitted signal, and the boundary with the underlying frozen debris is a clearly visible reflection (red dashed line), as is the boundary between the ice layer and the bedrock (blue dashed lines—see also the raw measurements in Figures A1 and A2 of Appendix A).

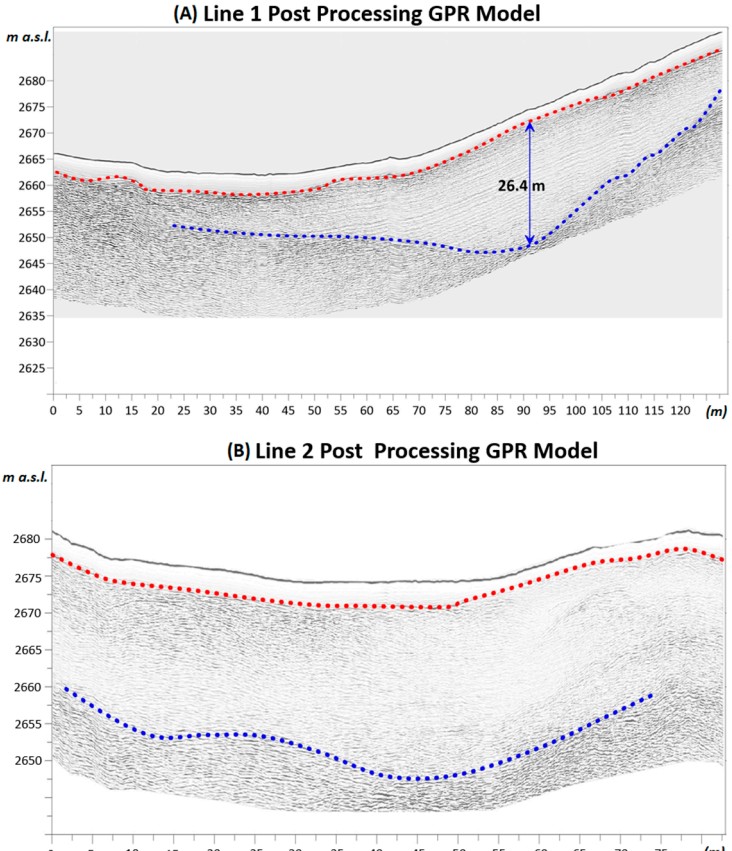

**Figure 4.** (**A**) Line 1 and (**B**) Line 2 post-processing GPR models. The red dashed line defines the boundary between the snow cover and the underlying frozen debris. The blue dashed line marks the limit between the ice layer and the bedrock; In (**A**), the blue arrow highlights the maximum thickness of the ice layer.

The maximum ice thickness value (26.4 m—blue arrow in Figure 4A) has been found along the longitudinal Line 1 at a distance of ≈90 m from the profile start. Along Line 2, the ice thicknesses do not show large variations; at the intersection point with Line 1, the thickness difference is practically negligible (<10%). Note that an important signal scattering occurs in the eastern part of the profile, suggesting that in this area, the ice layer has a larger presence of embedded debris and/or englacial water.

### 4.2. FDEM Inversion Results

Figure 5 shows the results of the FDEM inversion procedure (described in Section 3.3) applied to the field datasets acquired along Line 1 (Figure 5A) and Line 2 (Figure 5B). Both the inverted models have relatively low RRMSE values, 3.47% and 4.54%, respectively.

In Figure 6, the sensitivity of the measurements performed along Line 2 is presented (similar results were found for Line 1). Sensitivities are higher in the near subsurface and decrease to (approximately) zero at a depth of about 30 m. Consequently, we considered the uncertainty of the quasi-2D inverted conductivity models in the same way, and we defined the bottom of the sections at a depth of 30 m from the surface. It should be noted that the penetration depth of the survey is lower than that predicted by the instrument manufacturer (see Figure 2). This was expected since the FDEM investigation depth decreases in low electrical conductivity environments [11].

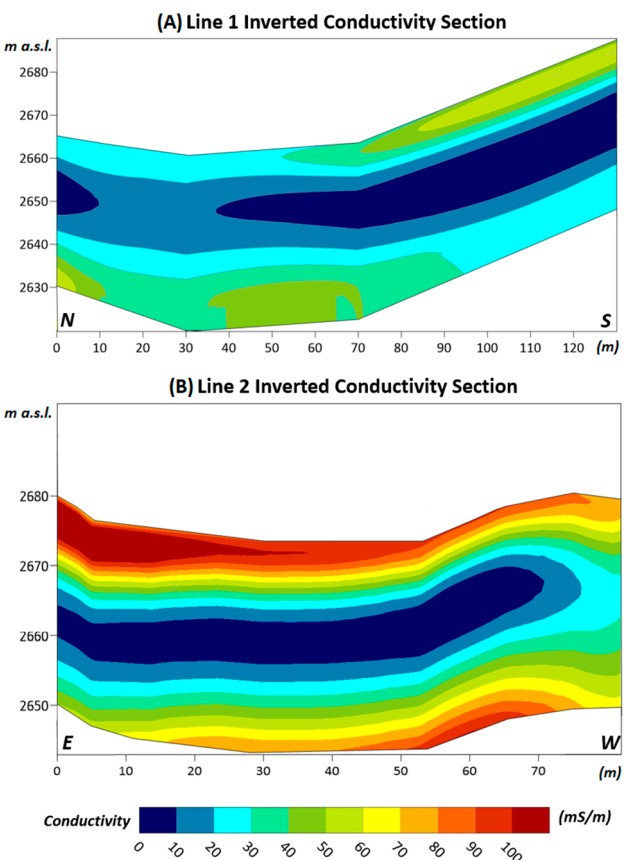

**Figure 5.** (**A**) Inverted conductivity model obtained with the dataset collected along Line 1; (**B**) inverted conductivity model obtained with the dataset acquired along Line 2. Note that, in the inversion process applied to the longitudinal profile Line 1, the dataset collected with coil separation *s* = 10 m and VCP mode has been removed due to a technical problem that occurred during acquisition.

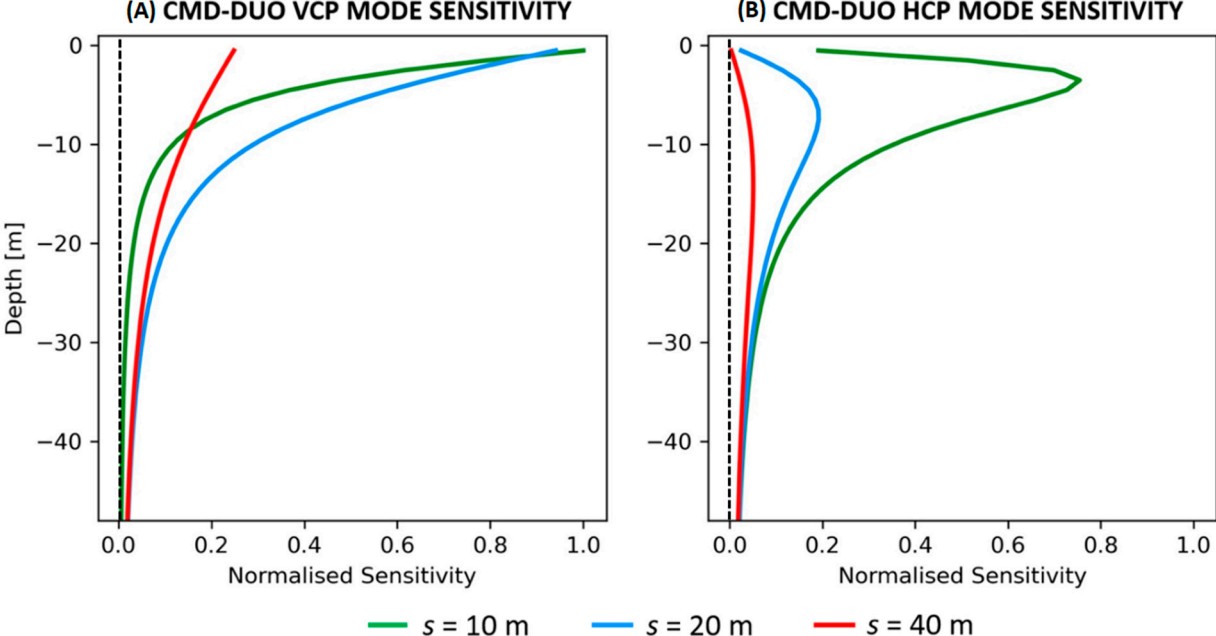

**Figure 6.** Normalized sensitivity pattern calculated for the FDEM measurements collected along Line 2 in (**A**) vertical coil position and (**B**) horizontal coil position. The black vertical dashed lines define the value of zero sensitivity for the acquired measurements.

From a structural point of view, the FDEM sections are very similar to their respective GPR models (see Figure 4A,B). For example, in Line 1 (Figure 5A), a clear low conductivity layer is definable in the middle of the section, from x ≈ 35 m until the end of the line, with the maximum thickness between x ≈ 90 m and x ≈ 100 m. Higher conductivity values are found in the uppermost layer and in the deeper area. In the same way, a three-layered structure can be defined within the resulting conductivity model of Line 2 (Figure 5B).

However, although the defined subsurface structures are very similar to their respective GPR models, the inverted electrical conductivity values are higher than expected. Therefore, a synthetic FDEM forward modelling process was computed to verify and evaluate the obtained results.

*4.3. FDEM Forward Modelling Results*

FDEM synthetic forward models were calculated to be compared with the results of the inverted FDEM field dataset. The longitudinal model (Figure 7A) was defined using information from the 2015 GPR survey [18]. Figure 7B shows the glacieret model of the orthogonal Line 2, based on the GPR survey results of March 2022.

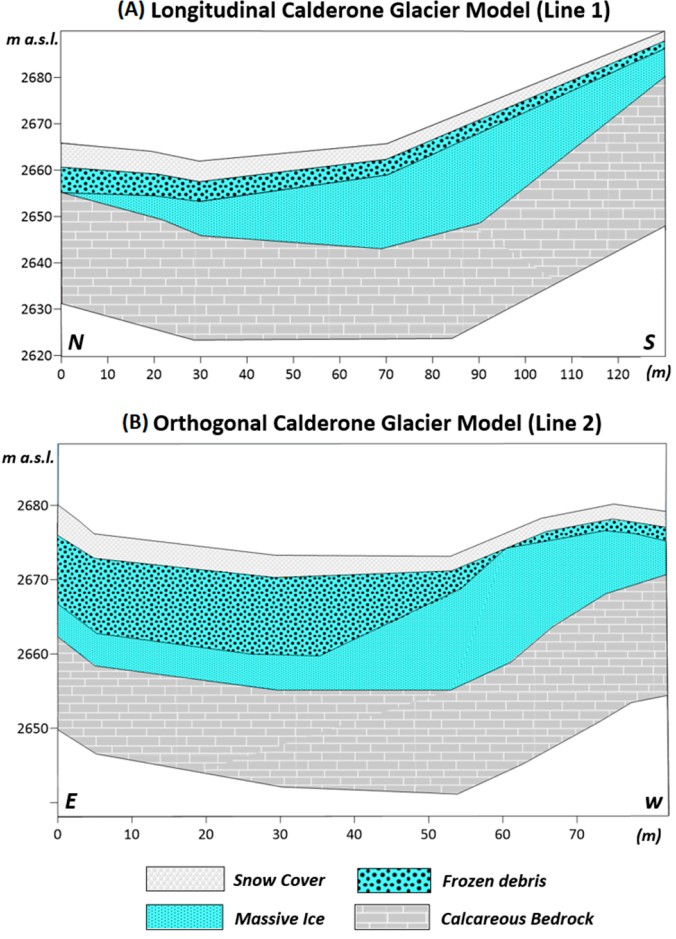

**Figure 7.** (**A**) A longitudinal model of the Calderone Glacieret as defined by Monaco and Scozzafava [18]. (**B**) An orthogonal Calderone Glacieret model as defined by the GPR surveys performed in March 2022.

These models have been used to perform the forward modelling process and to calculate FDEM synthetic datasets, simulating an apparatus with the same properties as the CMD-DUO instrument. In addition, the conductivity of each layer has been defined using both literature values and field measurements, as shown in Table 2.

**Table 2.** Electrical conductivity values from the literature used to perform the forward modeling process in the Calderone survey.

|  | Snow | Frozen Debris | Ice | Bedrock |
|---|---|---|---|---|
| **Conductivity (mS/m)** | 1 | $2 \times 10^{-2}$ | $1 \times 10^{-3}$ | $2 \times 10^{-1}$ |

The conductivity of the snow cover has been fixed to 1 mS/m according to the values measured by Pecci et al. [15] on the Calderone site. The frozen calcareous debris conductivity ($2 \times 10^{-2}$ mS/m) has been estimated considering the values found in calcareous rock glaciers by Pavoni et al. [24]. The ice of a temperate glacier practically acts as an electrical insulator and can be set at $1 \times 10^{-3}$ mS/m [11]. Finally, the calcareous bedrock conductivity has been evaluated as $2 \times 10^{-1}$ mS/m [28]. The FDEM synthetic datasets calculated with the forward modeling procedure were inverted using the same procedure as the real FDEM field data (see Section 3.3). Figure 8 shows the synthetic inverted conductivity models calculated for investigation in Line 1 (Figure 8A) and Line 2 (Figure 8B).

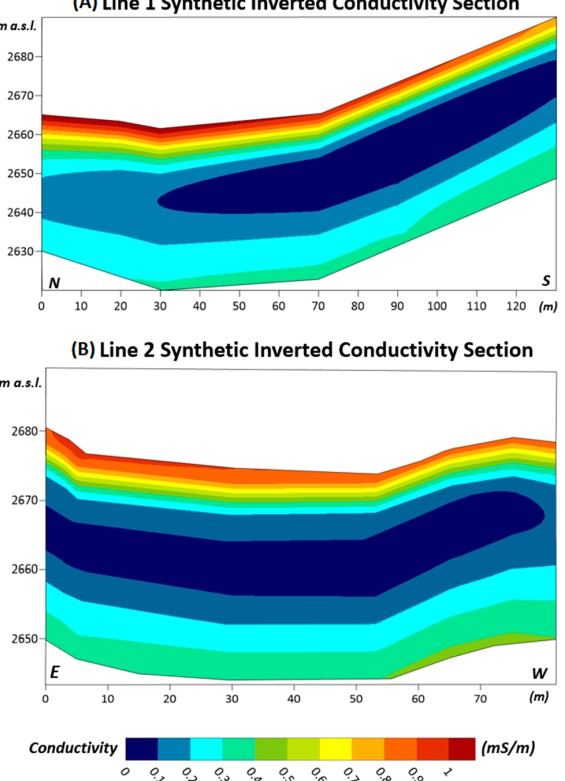

**Figure 8.** (**A**) Line 1 inverted conductivity sections of the synthetic datasets calculated with the forward modeling procedure applied to the longitudinal Calderone Glacieret model (Figure 7A) (**B**) Line 2 inverted conductivity sections of the synthetic datasets calculated with the forward modeling procedure applied to the orthogonal Calderone Glacieret model (Figure 7B).

Considering the results shown in Figure 8, we interpret values lower than $1 \times 10^{-1}$ mS/m as the ice-rich layer and values between $1 \times 10^{-1}$ and $2 \times 10^{-1}$ mS/m as an ice-debris mixture. Conductivity values higher than $2 \times 10^{-2}$ mS/m can be linked to unfrozen debris within the top layers and to bedrock at the bottom of the section. Values close to 1 mS/m may represent the upper snow cover layer. It can be noted that the subsurface structures defined with the synthetic FDEM models are very similar to the results found with our field datasets (see Figure 5), but the conductivity values are shifted by two orders of magnitude.

### 4.4. FDEM Correction Factor

The forward modelling procedure does not consider any instrumental resolution limit. Nevertheless, as mentioned before, the CMD-DUO has an instrumental limit of $1 \times 10^{-1}$ mS/m. Consequently, in the results of the inverted FDEM field dataset (Figure 5), we did not expect to find conductivity values in the same range as the synthetic models (from 0 to 1 mS/m—see Figure 8). Considering the results of the Line 1 GPR survey (Figure 4A), in the inverted model of Figure 5A, the conductivity boundary of the ice-rich layer can be set to $1 \times 10^{1}$ mS/m, and values lower than $2 \times 10^{2}$ mS/m can represent the ice-debris mixture. These values are two orders of magnitude higher than those found in the synthetic models (see Section 4.3 and Figure 8). This is practically the same difference existing between the instrumental resolution limit ($1 \times 10^{-1}$ mS/m) and the electrical conductivity of the ice in a temperate glacier ($1 \times 10^{-3}$ mS/m). Therefore, we applied a shifting correction factor of $1 \times 10^{-2}$ mS/m to the results of the inverted field dataset Line 1. This way, as it can be clearly seen in Figure 9A, the ice boundaries (ice-rich and ice-debris mixture) are represented by the same values defined in the synthetic models ($1 \times 10^{-1}$ mS/m and $2 \times 10^{-1}$ mS/m respectively). In Figure 9, the blue dashed line defines the boundary between the ice layer and the underlying bedrock, and the red dashed line is the boundary between the snow layer and frozen debris. The red star represents the position where the snow cover thickness (~5 m) was measured in March 2022 with a snow pit, and the yellow triangle is the location of the borehole drilled in April 2022. As shown in Figure 9C, the drilling detected an ice-debris mixture in the shallower part of the subsurface, followed by an ice-rich layer with a thickness of about 17 m. At the bottom of the ice-rich layer, debris was increased until the ice-rock boundary reached 27.2 m below ground level (without considering the snow cover thickness of about 1.5 m).

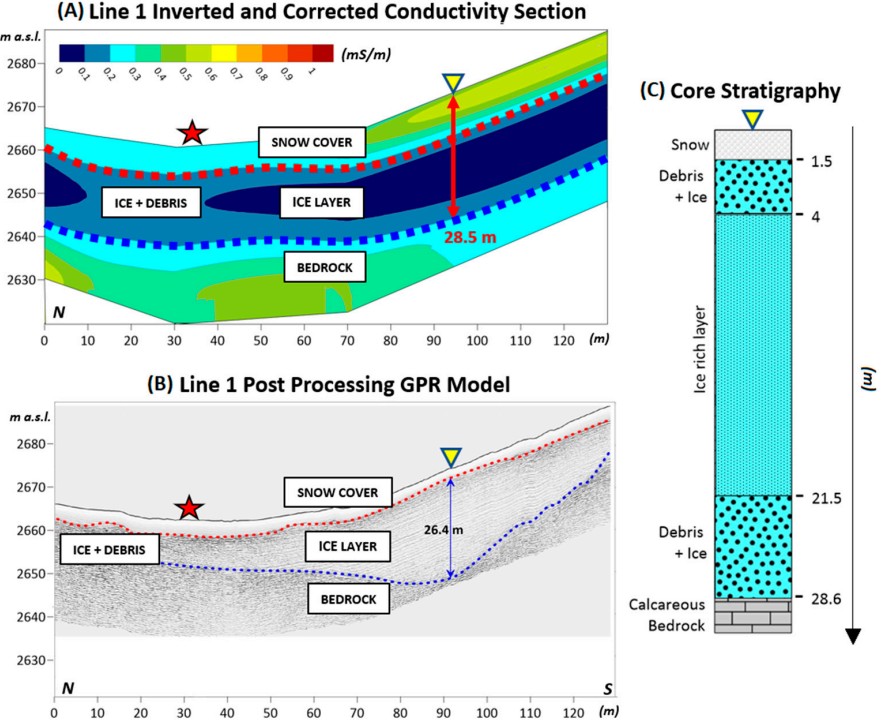

**Figure 9.** (**A**) Line 1 inverted and corrected conductivity section and (**B**) GPR model. Note that, in both the models, the boundaries between snow layer-frozen debris (red dashed line) and ice layer-bedrock (blue dashed line) are presented. The red star represents the position where the snow cover thickness was measured in March 2022, and the yellow triangle shows the location of the borehole drilled in April 2022. (**C**) Borehole stratigraphy defined during the extraction of the ice sample: 0–1.5 m snow cover, 1.5–4 m ice-debris mixture, 4–21.5 m ice-rich layer, 21.5–28.6 ice-debris mixture, and 28.6 m bedrock.

The same shifting correction factor has been applied to the result of the inverted FDEM field dataset Line 2. This way, the ice-rich layer boundary was set again to $1 \times 10^{-1}$ mS/m and the ice-debris mixture to $2 \times 10^{-1}$ mS/m. As for the investigation Line 1, the corrected conductivity section (Figure 10A) agrees with the glacieret structure defined by the corresponding GPR model (Figure 10B).

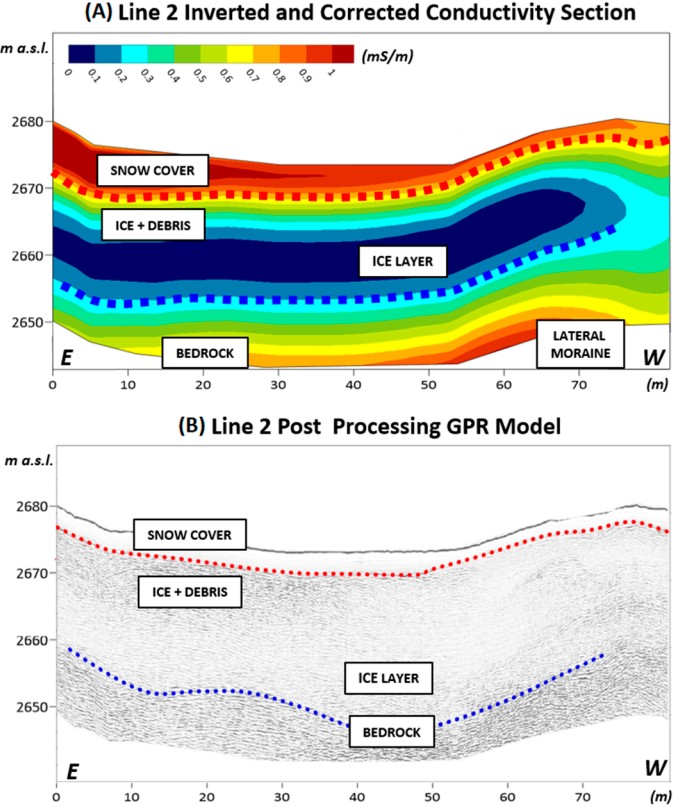

**Figure 10.** Line 2 (**A**) inverted and corrected conductivity section and (**B**) GPR model. Note that, in both the models, the boundaries between snow layer-frozen debris (red dashed line) and ice layer-bedrock (blue dashed line) are presented.

## 5. Discussion

The result of the FDEM inverted field dataset Line 1 (Figure 5A) suggests a subsurface structure very similar to the synthetic model (Figure 8A). Nevertheless, the conductivity scale is shifted by two orders of magnitude. This difference of magnitude is the same existing between the value of the instrumental resolution limit and the conductivity of the ice in a temperate glacier. The application of a fixed shifting correction factor of $1 \times 10^{-2}$ mS/m to the inverted field dataset allowed us to find conductivity values (Figures 9 and 10) in agreement with the synthetic ones (Figure 8). As in the rock glacier environments recently monitored by Pavoni et al. [24], the FDEM method does not aim to replicate the real electrical conductivities of the layers in the frozen subsurface, but to correctly estimate the subsurface structure thanks to a relative interpretation of the results (i.e., by defining areas with lower conductivities with possible ice fraction presence from areas without ice fraction). Reproducing the real conductivity values of the ice layers was out of the scope of the survey, taking into consideration the instrumental resolution limit. Moreover, the results of the FDEM forward modeling, which do not consider any instrumental limit, demonstrate that in these low conductive environments, we cannot retrieve the real conductivity values of the layers, even by applying the Maxwell full solution [21] in the inversion of synthetic datasets.

The subsurface structure suggested by the interpretation of the corrected FDEM conductivity sections is confirmed by the results of GPR surveys. The longitudinal GPR

profile Line 1 (Figure 4A) shows the negative trend of the Calderone Glacieret retreat. In fact, the ice-rich layer was easily identifiable along the entire longitudinal GPR profile measured in 2015 by Monaco and Scozzafava [18], but today seems to end at x ≈ 30 m. Therefore, in the last 7 years, between x = 0 m and x = 30 m of Line 1, a loss of massive ice may have occurred. This interpretation is confirmed by the inverted and corrected FDEM section (Figure 9A), where the ice-rich layer ($\sigma < 1 \times 10^{-1}$ mS/m) also disappears at x ≈ 30 m. In the middle layer, for x < 30 m, the conductivity values are between $1 \times 10^{-1} < \sigma < 2 \times 10^{-1}$ mS/m, suggesting the presence of ice but probably mixed with considerable quantities of debris and/or englacial water. In the GPR profile Line 1, the maximum thickness of the ice layer (26.4 m) can be placed at the distance of x ≈ 90 m, and the boundary ice layer bedrock is detected at a depth of 27.9 m. This information agrees with the FDEM section (Figure 9A), where the maximum thickness of the ice layer seems to be at a distance of x ≈ 90–100 m, and the boundary ice layer bedrock is defined at a depth of 28.5 m. The reliability of these results is confirmed by the stratigraphy defined during the drilling operations of April 2022 (see Figure 9C). The GPR model highlights a thinning of the ice layer towards the southern direction. On the other hand, in the conductivity model, the thickness variation is less evident, confirming the expected lower resolution of the FDEM technique compared to the GPR one. In the corrected FDEM section (Figure 9A), the layer representing the snow cover (conductivity values close to 1 mS/m as defined in the synthetic model, see Section 4.3 and Figure 8A) is missing. This is probably due to the absence of the dataset acquired in VCP mode and inter-coils distance *s* = 10 m, which detects the shallower layers during the measurements (see Figure 2). This dataset was removed because we had technical problems during the acquisition. On the other hand, in the GPR model (Figure 4A), the thickness variation of the snow cover layer is clearly visible moving from the south to the north direction (see also Figure A1 Appendix A). In the southern area, the snow cover is a couple of meters deep, as confirmed by the drilling (Figure 9C), while towards the glacieret front (north), it tends to increase up to 5 m, as also measured during the field operations with a snow pit (red star Figures 1A and 9A,B). A similar trend is also found in the GPR profile Line 2 (Figure 4B). The snow layer has a greater thickness in the east direction and thins out towards the west (see also Figure A2 Appendix A). In this case, the variation is also detected by the corrected FDEM section (Figure 10A), where the dataset VCP *s* = 10 m was correctly acquired. The GPR profile Line 2 confirms the presence of the ice layer but with a maximum thickness slightly lower than that found for the longitudinal profile. This information agrees with the trend defined by the results of Line 1, where the maximum thickness of the ice layer is found at x ≈ 90 m, and it is thinner in both northern and southern directions. Along Line 2, the ice thickness is greater in the western direction of the profile (50 < x < 60 m) and tends to thin towards the east, as confirmed by the corrected conductivity section (Figure 10A). Note that, in comparison to the synthetic FDEM model (Figure 7B), the bottom of the corrected conductivity section has higher values than expected for the bedrock, particularly in the western direction. In this area, probably, the ice layer is not directly in contact with the calcareous bedrock but instead with the lateral moraine, which typically has higher conductivity values than the bedrock. This information cannot be defined in the radargram, which on the other hand, allows the detection of the thickness of the different layers with much more precision. Therefore, integrating the GPR survey with the FDEM method can help improve the characterization of the glacieret structure and better define the composition of the different layers. As for the GPR, the acquisition of the FDEM data is relatively simple as the measurements are taken continuously by moving with the two probes along the survey line. Acquisition of both GPR and FDEM surveys is then relatively fast. On the other hand, the analysis of the FDEM data requires particular attention, and the instrumental limit resolution must be considered. Performing forward modeling is advisable for a reliable interpretation of the field dataset inversion results. Furthermore, to facilitate the inversion routine, it is also useful to filter and smooth the dataset. Using the CS function and the Gauss-Newton optimization method allows us to reduce the computational effort of the inversion

process, and to obtain a quasi-2D conductivity model of the subsurface easily. It is also possible to apply more complicated inversion methods based on Maxwell's laws. However, as discussed by McLachlan et al. [21], in low electrical conductivity environments, the differences between the models obtained with the CS function and the other methods are minimized.

## 6. Conclusions

The results of the geophysical investigations performed on the Calderone Glacieret confirm the excellent capabilities of the GPR method to characterize glacial environments. The measurements, acquired with a modern 200 MHz monostatic digital antenna, define with extreme precision the thickness of the snowpack and the boundary between the ice layer and the calcareous bedrock. Future development for the GPR measurements may be to apply the method proposed by Santin et al. [29] to estimate the debris content within the layers composed of an ice-debris mixture. Furthermore, in the case of periodic measurements in time-lapse configuration, this method could help to estimate possible variation (thickness and lateral termination) of the ice layer in the Calderone Glacieret in the next future.

The results obtained with the separated-coils FDEM device on the Calderone Glacieret show the potential of this technique to be applied even in a low-conductive environment. Therefore, the FDEM method can be integrated into the structural characterization of a subsurface containing ice layers. In our study case, the obtained FDEM conductivity models, combined with the GPR models, were successfully used by the Ice Memory project team. Thanks to the geophysical surveys, they defined the drilling position on the Calderone Glacieret, where the ice layer was presumed to be thicker. Furthermore, the extracted cores proved the reliability of the applied geophysical method, confirming the structures of the subsurface layers. Considering these promising results, the future perspective is to use the FDEM separated-coils device also in rock glacier environments. In these periglacial landforms, the GPR technique is, in fact, more complicated to apply since the coarse-blocky surface hinders data acquisition and enhances the problem of signal scattering. However, the FDEM method is not affected by these problems and does not need good galvanic contact with the surface as required by the ERT method [30]. Moreover, the logistic effort of the FDEM investigation is much lower in comparison to the ERT surveys and can represent a reliable preliminary investigation tool to evaluate the subsoil structure.

The FDEM technique is not here proposed as a substitute for the GPR method in glacier or glacieret environments since the latter remains the best in terms of resolution capabilities. FDEM measurements can rather represent a convenient integration to other surveys to support a better reconstruction of the subsurface structure and composition.

**Author Contributions:** All the authors have been involved in data acquisition; M.P. performed the data processing of the FDEM method; S.U. performed the data processing of the GPR method. All authors have read and agreed to the published version of the manuscript.

**Funding:** This research was funded by FISR Ice Memory, CIPE n. 78 (7 August 2017) published in G.U. n. 277 (27 November 2017).

**Data Availability Statement:** The datasets used to obtain the results presented in this work are available at the open-source repository: https://zenodo.org/badge/latestdoi/611632045 (accessed on 9 March 2023). (DOI: 10.5281/zenodo.7711391).

**Acknowledgments:** The Authors thank Massimo Pecci for the relevant discussion about the Calderone Glacieret history, the National Fire Department (Corpo Nazionale dei Vigili del Fuoco) for the logistic helicopter support, Pinuccio D'Aquila (Engineering Srls) for the photogrammetric acquisition survey of Calderone site (Figure 1A), the photographer Riccardo Selvatico for the images presented in Figure 1B,C, and the mountain guides Paolo Conz and Thomas Ballerin for the field support. Finally, the authors thank Tegan Blount for the review of the English language, and two anonymous reviewers of the manuscript whose comments and suggestions helped to improve the quality of our work.

**Conflicts of Interest:** The authors declare no conflict of interest.

**Appendix A**

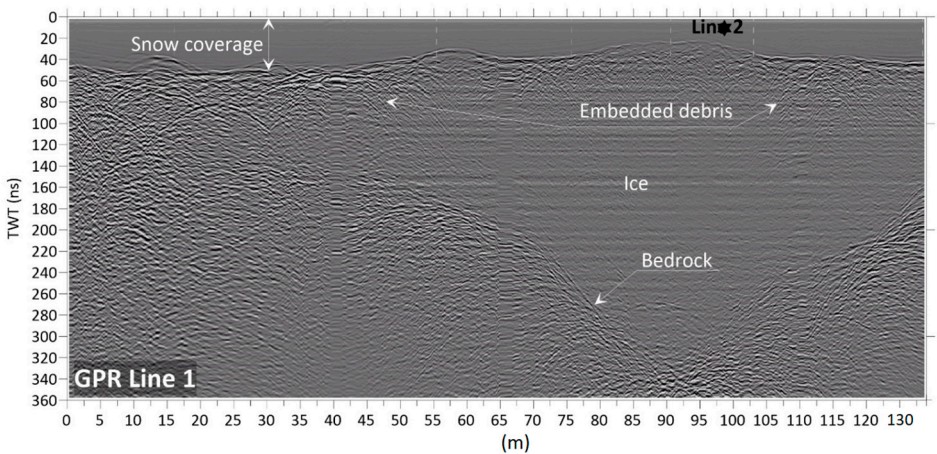

**Figure A1.** Interpretation of the GPR model Line 1 pre-processing.

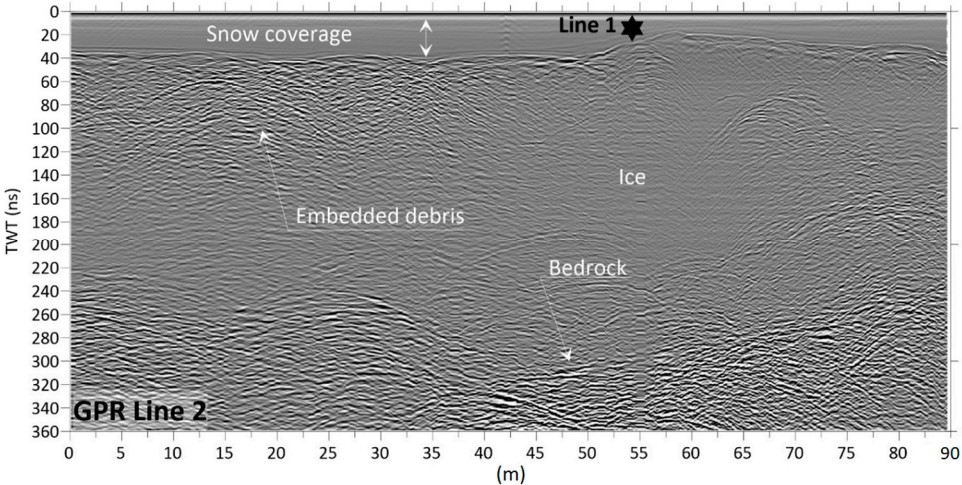

**Figure A2.** Interpretation of the GPR model Line 2 pre-processing.

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
