# Peer review of "Combining Ground Penetrating Radar and Frequency Domain Electromagnetic Surveys to Characterize the Structure of the Calderone Glacieret (Gran Sasso d’Italia, Italy)"

_remotesensing, doi:10.3390/rs15102615_

Round 1

Reviewer 1 Report

Dear authors,

Your manuscript presents the results of geophysical surveys on the Calderone Glacier and interprets the obtained results in terms of the internal structure of the glacier. Currently, FDEM surveys are an ongoing and expanding research niche, and papers that demonstrate the benefits of this technique are highly valuable. However, the study has conceptual and formatting errors (paragraphs and sections need to be rewritten) that make this paper unsuitable for publication in its current form (major revision).

The authors could also benefit from reading more relevant literature on small glaciers (glacierets) and geophysical techniques applied to glacier ice.

General comments:

Background:

There is some misunderstanding about your study area and the "Calderone Glacier". A glacier is defined as surface ice that shows clear signs of flow - especially crevasses - from an accumulation area to an ablation area. This requires a minimum size - several hundred meters in length and elevation range, or 0.1 km2 in area (see Glacier Inventories). Due to negative mass balance and retreat, Calderone Glacier is no longer a glacier in the strict sense. It should be referred to as a glacieret (ice body smaller than 0.1 km2), which is actually the remnant of a fragmented glacier (an upper and lower glacieret that are currently separated).

Introduction:

The introduction is currently a very long paragraph. This should be restructured into at least two or three paragraphs with state of the art, background information, and objectives of the study.

Discussion:

The discussion looks quite tangled with different terms and should be organized into two or more central ideas. I would suggest adding a discussion of uncertainty in your data.

Conclusions:

Here I am missing one of the motivations for your work on ice coring and sample preservation. So the usefulness of your new datasets should be stated in relation to the ICE memory program.

Please see the attached pdf file for a detailed review of the manuscript.

The paper is not well organized and would benefit from assistance with standard academic English rules. The document would benefit from a restructuring of its sections. There is no clear distinction between introduction, study site, and methods. There are some words and phrases that I found quite cryptic (see annotated pdf file). Also, some of the references cited are odd and unrelated.

Author Response

Dear Reviewer 1, thank you for your comments that clearly improved our manuscript. Here attached you can find a file docx with a point-by-point response to the suggestions/comments. 

Reviewer 2 Report

Moderate editing of English language

Author Response

Dear Reviewer 2, thank you for your comments that clearly improved our manuscript. Here attached you can find a file docx with a point-by-point response to the comments/suggestions.

Round 2

Reviewer 1 Report

Dear authors,

Thank you for providing an improved manuscript version. However, there are still a few issues with this document. While the changes you have made are commendable, a few additional areas require further attention to meet publication standards.

  1. The introduction is still a long paraphrase that should be broken into two or three paragraphs englobing the research's key ideas, concepts and objectives. The main body should then contain a detailed analysis of the research, which should be divided into logical sections and subsections. Finally, the conclusion should be a summary of the main findings.

  1. There are still inconsistencies regarding the terminology "Glacier". Please check the text and change it to glacieret or ice bodies. Additionally, please highlight the differences between a glacier and a glacieret. Finally, please check the text for any other inconsistencies in terms of terminology.

Best regards.

There has been a considerable improvement in the quality of the English, but it still needs some polishing.

Author Response

Reviewer #1

Comment: Thank you for providing an improved manuscript version. However, there are still a few issues with this document. While the changes you have made are commendable, a few additional areas require further attention to meet publication standards:

  • The introduction is still a long paraphrase that should be broken into two or three paragraphs englobing the research's key ideas, concepts and objectives. The main body should then contain a detailed analysis of the research, which should be divided into logical sections and subsections. Finally, the conclusion should be a summary of the main findings.

REPLY: We thank the Anonymous Reviewer for the suggestion and the introduction has been modified and divided into different paragraphs as suggested (see revised manuscript).

We respect the suggestion of the Anonymous Reviewer, but we followed the author's guidelines of Remote Sensing for the structure of the paper: 1. Introduction, 2. Site Description, 3. Methods (3.1 Ground Penetrating Radar (GPR), 3.2 Frequency Domain Electro Magnetic (FDEM), and 3.3 FDEM forward and inverse modeling), 4. Results (4.1 GPR results, 4.2 FDEM inversion results, 4.3 FDEM forward modeling results, and FDEM correction factor), 5. Discussion, and 6. Conclusions.  Our form meets the requirements of the MDPI editorial office. We widely discussed the results and the findings in the Discussion section, and we added the Conclusion chapter (even if not requested by RS, see below the guideline) just to insert a few considerations and possible future development of our studies.

The RS guideline, Conclusions: This section is not mandatory but can be added to the manuscript if the discussion is unusually long or complex

  • There are still inconsistencies regarding the terminology "Glacier". Please check the text and change it to glacieret or ice bodies. Additionally, please highlight the differences between a glacier and a glacieret. Finally, please check the text for any other inconsistencies in terms of terminology.

REPLY: We thank the Anonymous Reviewer for the indication and we modified the text of the Manuscript avoiding any reference to Calderone Glacier (see revised Manuscript). We better highlighted the difference between glacier and glacieret in the introduction (see lines 31-33 revised Manuscript).  Another round of English editing was also done.

Author comment: We really thank the first Reviewer for his/her comments and suggestions that helped to improve the quality of our manuscript.

Best regards.

Reviewer 2 Report

Thank you very much to the authors for the care given to the revision of this manuscript and to have consider my suggestions and comments. It was a real pleasure to read this new version. Please have a last global reading, I have seen some typo errors:

L38: check the sentence

L78: above sea level (a.s.l.) ...

L254: ...this sentence has no meaning, as the title of the figure already indicates

L357: Figure 9C...

L410: at a distance of

L500: thank...

Moderate editing of English language

Author Response

Reviewer #2

Comment: Thank you very much to the authors for the care given to the revision of this manuscript and to have consider my suggestions and comments. It was a real pleasure to read this new version. Please have a last global reading, I have seen some typo errors:

  • L38: check the sentence

REPLY: we modified the sentence in lines 37-38-39 (see revised manuscript).

  • L78: above sea level (a.s.l.) ...

REPLY: we modified the sentence in line 78 (see revised manuscript).

  • L254: ...this sentence has no meaning, as the title of the figure already indicates.

REPLY: this sentence has been inserted to present the following Figure 4. We think that is better to present a figure (or a table) in the main text before showing that figure (or table). Nevertheless, we agree with the Reviewer that the sentence of L254 alone before the figure is not appropriate, therefore we modified the text in the revised manuscript. 

  • L357: Figure 9C...

REPLY: sorry for the error, we did the correction in the revised manuscript.

  • L410: at a distance of

REPLY: sorry for the error, we did the correction in the revised manuscript.

  • L500: thank...

REPLY: sorry for the error, we did the correction in the revised manuscript.

Another round of English editing was also done.

Author comment: We really thank the second Reviewer for his/her comments and suggestions that helped to improve the quality of our manuscript.

Best regards.
